# Fibroblastic Connective Tissue Nevus Mimicking Lipoma on Ultrasound: Case Report and Brief Review

Paola Lara-Valencia *  , Anna Ybo and Anna Coter

Unit of Pathology, Department of Immunology, Genetics and Pathology, Uppsala University Hospital,
Uppsala University, 75237 Uppsala, Sweden; anna.ybo@regionvastmanland.se (A.Y.);
anna.coter@regionvastmanland.se (A.C.)
* Correspondence: paola.lara.valencia@igp.uu.se

**Abstract:** Fibroblastic connective tissue nevus (FCTN) is a rare, benign, and recently described dermal mesenchymal lesion characterized by CD34-positive spindle cells. We present a case of FCTN on the upper back of a 9-month-old boy who was diagnosed with a benign lipoma by ultrasound.

**Keywords:** CD34; mesenchymal skin lesion; lipoma; dermatofibrosarcoma protuberans

## 1. Introduction

Fibroblastic connective tissue nevus (FCTN) is a rare, cutaneous mesenchymal lesion initially described by de Feraudy and Fletcher in 2012 [1]. The clinical features of FCTN are variable, usually presenting as a slow-growing, single, and asymptomatic lesion. Histologically, the lesion shows a deep dermal proliferation of fibroblastic cells arranged in disordered short fascicles entrapping collagen bundles and sweat glands without destruction of the structures; these cells are CD34-positive.

## 2. Case Report

A healthy 9-month-old male presented with an asymptomatic, well-defined, nonmobile subcutaneous nodule of the upper back. The lesion appeared in the first two months of life and slowly increased in size. The ultrasound showed a 4-cm long heterogenous lesion with a solid appearance and an echogenicity similar to that of fat. There was no connection with the deep fascia. The ultrasound diagnosis was compatible with a lipoma and was surgically removed as the lesion had grown in the last month. Since clinically and ultrasonographically the diagnosis was unclear, it was preferential to remove it.

Histopathological examination revealed a papillomatous epidermis without atypia. In the deep reticular dermis and superficial subcutis, a poorly circumscribed lesion consisting of bundles of monomorphic spindle cells with pale eosinophilic cytoplasm arranged in a disorderly manner in the short fascicle in the stroma, was observed. Furthermore, entrapped adnexal structures with abundant mature adipocytic components were also found. Atypical mitoses, cytologic atypia, or other features suggestive of malignant transformation were absent. Immunohistochemistry revealed that the cells were CD34-positive, while they were negative for smooth muscle actin, S100, desmin, myoglobin, Factor XIIIa, and CD31 (Figure 1). Moreover, the elastic fibers were fragmented. Therefore, the definitive diagnosis was fibroblastic connective tissue nevus.

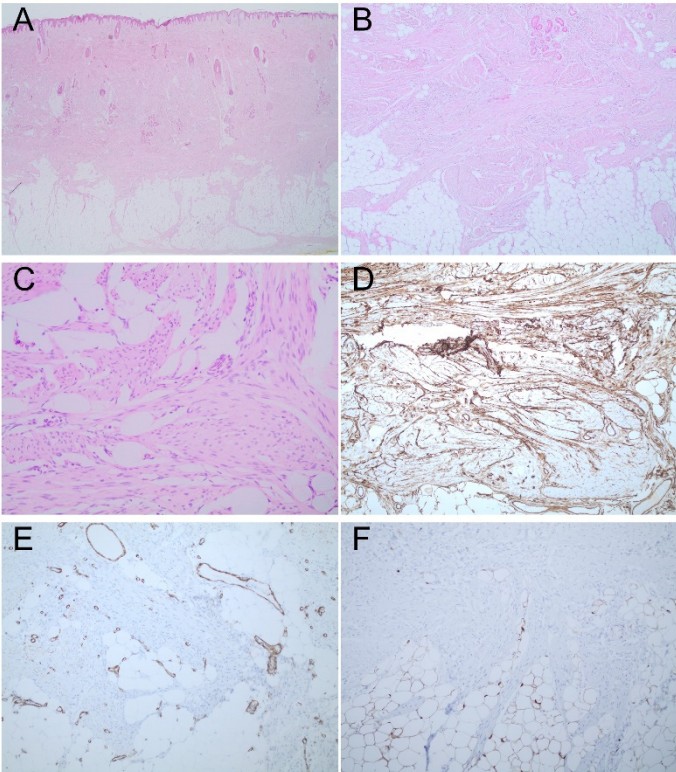

**Figure 1.** Fibroblastic connective tissue nevus mimicking a lipoma. (**A**) Poorly circumscribed lesion of spindle cells in the deep dermis (hematoxylin and eosin, original magnification 4×). (**B**) Entrap adnexal structures with superficial mature adipocytic components (hematoxylin and eosin, original magnification 10×). (**C**) Bundles of monomorphic spindle cells with pale eosinophilic cytoplasm arranged in a disorderly manner (hematoxylin and eosin, original magnification 20×). (**D**) Diffuse positive CD34 immunostaining. (**E**) Negative immunostaining for smooth muscle actin. (**F**) Negative immunostaining for S100.

### 3. Discussion

Fibroblastic connective tissue nevus is an infrequent, benign, cutaneous mesenchymal lesion of fibroblastic and myofibroblastic origin, first described by de Feraudy and Fletcher [1]. The clinical features of FCTN are variable, usually presenting as a slow-growing, single, and asymptomatic lesion, which may be plaque-like, or as a cutaneous nodule with an indolent course and may show partial spontaneous regression [2]. This kind of tumor is generally acquired and is usually predominant in young girls (median age of presentation 10 years) [1] while congenital FCTN has rarely been reported [3,4], There are less than 50 reported cases, and the incidence is unknown [3]. Most FCTN cases predominantly appear on the head and neck, trunk, and limbs, with a variable size from 0.3 to 4 cm in dimension [1,2]. No syndromic association has been seen.

Microscopically, the lesion shows a poorly circumscribed, deep dermal proliferation of fibroblastic cells arranged in disordered short fascicles entrapping collagen bundles and sweat glands without destruction of the structures. There is no evidence of cytological atypia, mitosis, or necrosis. Immunohistochemically, these cells are CD34-positive in 87% of cases, smooth muscle actin-positive in less than 50% of the cases, but S100-negative in all cases, which is consistent with a fibroblastic/myofibroblastic process [1].

The differential diagnosis is broad and includes dermatofibroma, dermatomyofibroma, plaque-stage dermatofibrosarcoma protuberans (DFSP), CD34-positive plaque-like dermal fibroma, fibroblastic-predominant plexiform fibrohistiocytic tumor, and fibrous hamartoma of infancy, among others [1–6]. Furthermore, a congenital CD34-positive dermohypodermal spindle-cell neoplasm harboring a novel KHDRBS1–NTRK3 fusion has been described as

a possible new subgroup of pediatric cutaneous spindle-cell neoplasms that can potentially mimic FCTN [5].

One of the most important differential diagnoses includes DFSP [7] which may be difficult with a superficial punch biopsy. DFSP is a tumor of intermediate malignancy of slow growth, is locally aggressive, and has a tendency to reappear locally unless completely removed surgically and may undergo a higher-grade fibrosarcomatous transformation, which has metastatic potential [8]. Histologically, it is characterized by a proliferation of spindle-shaped atypical CD34-positive cells and can sometimes be confused with the pattern of FCTN. In some cases, it is necessary to perform cytogenetic translocation studies. Approximately 90% of DFSP cases have a COL1A1–PDGFB translocation, which is absent in FCTN [9].

## 4. Conclusions

In summary, we report a case of an infrequent fibroblastic connective tissue nevus with an ultrasonographic appearance of a lipoma. The histological and immunohistochemical features support our diagnosis of FCTN. This lesion is benign, and patients have a good prognosis with local excision.

**Author Contributions:** Con-ceptualization, P.L.-V., A.Y. and A.C.; methodology, P.L.-V.; software, P.L.-V.; writing—original draft preparation, P.L.-V., A.Y. and A.C.; writing—review and editing, P.L.-V. All authors have read and agreed to the published version of the manuscript.

**Funding:** The authors received no financial support for the research, authorship, and/or publication of this article.

**Institutional Review Board Statement:** Not applicable.

**Informed Consent Statement:** Not applicable.

**Data Availability Statement:** Not applicable.

**Conflicts of Interest:** The authors declare no conflict of interest.

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
