# Peer review of "Fibroblastic Connective Tissue Nevus Mimicking Lipoma on Ultrasound: Case Report and Brief Review"

_dermatopathology, doi:10.3390/dermatopathology9010005_

Round 1

Reviewer 1 Report

Article describes a rare case of congenital Fibroblastic connective tissue nevus (FCTN): however interesting there are some incorrect statements: 1) the ultrasound and clinical appearance (the lesion is described as non-mobile) are not suggestive of lipoma

2) if the diagnosis of lipoma is posed, however, it is not explained why to remove it (under general anesthesia ????)

3) I think for these reasons it would be better to indicate that "since clinically and ultrasonographically the diagnosis was highly doubtful it was preferred to remove it.( with citation of :Fabiano A, Moro R, Zane C, Caravello S, Fusano M, Calzavara-Pinton P, Gualdi G. Pediatric dermatologic surgery: our experience. G Ital Dermatol Venereol. 2020 Dec;155(6):775-779. doi: 10.23736/S0392-0488.18.06140-0. Epub 2018 Sep 24. PMID: 30251807.)

Author Response

Dear reviewer,

We appreciate your comments and suggestions and having considered our article for publication in the journal Dermatopathology. Below we answer your questions and comments, which we will add to our article.

  1. The ultrasound and clinical appearance (the lesion is described as non-mobile) are not suggestive of lipoma.

The lesion was non-mobile however there was not connection with the deep fascia. The diagnosis was performed by ultrasound that showed a heterogenous lesion with a solid appearance and relatively an echogenicity similar to that of fat.

  1. if the diagnosis of lipoma is posed, however, it is not explained why to remove it (under general anesthesia ????)

The lesion was removed for having grown in the last year and it was removed under general anesthesia.

  1. I think for these reasons it would be better to indicate that "since clinically and ultrasonographically the diagnosis was highly doubtful it was preferred to remove it.( with citation of :Fabiano A, Moro R, Zane C, Caravello S, Fusano M, Calzavara-Pinton P, Gualdi G. Pediatric dermatologic surgery: our experience. G Ital Dermatol Venereol. 2020 Dec;155(6):775-779. doi: 10.23736/S0392-0488.18.06140-0. Epub 2018 Sep 24. PMID: 30251807.)

Reviewer 2 Report

Dear Author,

Thank you for this interesting case about fibroblastic connective tissue nevus mimicking lipoma on ultrasound.

  • The discussion is not clearly structured and we ask to better discuss the case from a clinical point of view.
  • Please, would you specify if dermoscopy was performed? If not, why?
  • We would suggest avoiding unnecessary repetitions (Lines 45-46).
  • Please would you explain what does “is involves” means? (Line 67)
  • We suggest to better discuss what “Proper diagnosis is important because it determines the choice of treatment for this lesion” means, explaining current therapies available for this condition (Line 75)
  • Image 1: the color of histopathology slides is faded, and image quality low. Would you mind improving image contrast/quality ?

Author Response

Dear reviewer,

We appreciate your comments and suggestions and having considered our article for publication in the journal Dermatopathology. Below we answer your questions and comments, which we will add to our article.

  • The discussion is not clearly structured and we ask to better discuss the case from a clinical point of view.

We will do a better clinical discussion.

  • Please, would you specify if dermoscopy was performed? If not, why?

The patient was managed by pediatric surgery and not by dermatology, which is why dermpscopy was not performed.

  • We would suggest avoiding unnecessary repetitions (Lines 45-46).

Thank you, we will eliminate the line.

  • Please would you explain what does “is involves” means? (Line 67)

“is involves” means that the proliferation is characterized by …. We will change the sentence.

  • We suggest to better discuss what “Proper diagnosis is important because it determines the choice of treatment for this lesion” means, explaining current therapies available for this condition (Line 75)

We will improve the discussion.

  • Image 1: the color of histopathology slides is faded, and image quality low. Would you mind improving image contrast/quality ?

We will try to improve the color of histopathology images, however the H-E is weak. Unfortunately the artwork was not ready today 11/01/2022. I can get the images 12/01/2022, could we send the images tomorrow?
